# Vitamin C as Scavenger of Reactive Oxygen Species during Healing after Myocardial Infarction

**DOI:** 10.3390/ijms25063114

**Published:** 2024-03-07

**Authors:** Huabo Zheng, Yichen Xu, Elisa A. Liehn, Mihaela Rusu

**Affiliations:** 1Department of Cardiology, Angiology and Intensive Care, University Hospital, Rheinisch-Westfälische Technische Hochschule Aachen University, 52074 Aachen, Germany; zhenghuabo129@outlook.com; 2Institute of Molecular Medicine, University of Southern Denmark, Campusvej 55, 5230 Odense, Denmark; yixu@ukaachen.de; 3Department of Histology and Embryology, Medicine and Life Sciences, Hainan Medical University, Haikou 571199, China; 4National Institute of Pathology “Victor Babes”, Splaiul Independentei Nr. 99-101, 050096 Bucharest, Romania; 5Institute of Applied Medical Engineering, Helmholtz Institute, Medical Faculty, Rheinisch-Westfälische Technische Hochschule Aachen University, 52074 Aachen, Germany

**Keywords:** vitamin C supplementation, reactive oxidative species (ROS), antioxidant capacity, oxidative stress, preventive strategy, cardiovascular diseases, myocardial infarction

## Abstract

Currently, coronary artery bypass and reperfusion therapies are considered the gold standard in long-term treatments to restore heart function after acute myocardial infarction. As a drawback of these restoring strategies, reperfusion after an ischemic insult and sudden oxygen exposure lead to the exacerbated synthesis of additional reactive oxidative species and the persistence of increased oxidation levels. Attempts based on antioxidant treatment have failed to achieve an effective therapy for cardiovascular disease patients. The controversial use of vitamin C as an antioxidant in clinical practice is comprehensively systematized and discussed in this review. The dose-dependent adsorption and release kinetics mechanism of vitamin C is complex; however, this review may provide a holistic perspective on its potential as a preventive supplement and/or for combined precise and targeted therapeutics in cardiovascular management therapy.

## 1. Introduction

Myocardial infarction (MI) is one of the most severe clinical manifestations of heart disease, which is a leading cause of death worldwide [1] despite considerable clinical management improvements in the past decade [2,3]. The first recognized cause of MI is the chemo-mechanical disruption of vulnerable or unstable plaque, which leads to the formation of a thrombus and occlusion of the vessels [4]. Mechanical (balloon angioplasty) and pharmacological (thrombolysis) modern management therapies restore blood flow in patients with MI [5,6,7,8] by removing the plaque or occluded thrombus from the blocked coronary artery [9]. While MI leads to the recruitment and activation of inflammatory cells (e.g., neutrophils [10], macrophages [11], monocytes [12], regulatory T-cells [13], CD4^+^ T-cells [14], mast cells [15]), reperfusion results in a burst of free radical formation [16]. Thus, the increased oxidative stress and imbalanced levels of the production and accumulation of reactive oxygen species (ROS) will exaggerate tissular damage, known as reperfusion injury, which results in increased myocyte damage and later increased fibrosis and remodeling [17,18]. Despite tremendous progress towards reducing post-ischemic reperfusion events, patients undergoing MI still experience increased mortality and morbidity [2,3]. Understanding changes in physio-pathological parameters, including heart function [19], heart frequency [20], and vascular integrity [21], is important for establishing novel therapeutic strategies to prevent or minimize heart failure.

A key component of cardiomyocyte death is mitochondrial dysfunction during ischemic reperfusion injury [22]. At rest, the energy reservoir produced by mitochondrial activity represents 6–30 kg of adenosine triphosphate (ATP) daily [23]. This provides the heart with the amount of ATP necessary for a cardiac muscle demand of 100,000 beats daily [24]. Although the metabolism of the heart appears to be complex, studies show that three main mechanisms are highly involved in contraction: (i) glycolysis and β-oxidation of free fatty acids are the primary fuel sources; (ii) oxidative phosphorylation of adenosine diphosphate (ADP) will produce ATP; and (iii) high-energy transfer from ATP to the heart occurs via myofibrils.

Antioxidant therapies have been proposed to efficiently reduce oxidative stress and cellular damage during MI by boosting the body’s natural defense mechanisms. Polyphenols are known to have a broad spectrum of antioxidant benefits [25]. Vitamin E is particularly important in protecting cell membranes from oxidative damage [26]. Vitamin C is effective in neutralizing ‘harmful’ free radicals [27] and, when combined with vitamin E, it acts synergistically to reinforce the protective effects [28]. Moreover, the effects of vitamin C extend beyond the mere neutralization of free radicals [29,30]. By donating its electrons, vitamin C can protect important biomolecules (proteins, lipids, carbohydrates, and nucleic acids) from being damaged by oxidants, including ROS [31,32]

Previous studies of the association between vitamin C levels and MI seem to have conflicting results. Generally, fruit and vegetable intake is known to be reflected in plasma vitamin C and is inversely associated with the risk of heart failure [33]. The multivariant Cox model of a 6-year stratified cohort study (age, season, and examination year) showed that patients with a low plasma vitamin C concentration (<11.4 μM) had a significant association (3.5-fold) with an increased risk of MI [34]. Another study involving 180 male patients could not find any correlation related to MI risk, irrespective of smoking status [35]. Similarly, other studies have failed to demonstrate the beneficial effects of vitamin C intake on cardiovascular health and other complications [36], including MI [37,38]. However, a prospective study of 3919 older men followed up for a mean period of 11 years demonstrated the relationship between a reduced risk of heart failure and higher plasma vitamin C levels, independent of MI [39].

This review comprehensively revisits the role of vitamin C as an antioxidant during healing after MI, revealing its mechanisms and offering prospects for clinical use. 

## 2. Metabolism of Vitamin C

Vitamin C is an important co-enzymatic factor for cellular function and survival. In vivo or in vitro, vitamin C undergoes oxidation to dehydroascorbic acid (DHA), which is promoted by oxidants like hydrogen peroxide (H_2_O_2_) or catalyzed by the enzyme vitamin C oxidase [40,41]. In vivo, DHA has a half-life of only a few minutes, and it will normally be reduced back to vitamin C by thioredoxin reductase, dehydroascorbate oxoreductase, and 3-α-hydroxysteroid dehydrogenase [41,42]. In the liver and the kidney, dehydroascorbic acid will be hydrolyzed to 2,3-diketo-L-gulonate and then decarboxylated to L-xylonate and L-lyxonate [43,44], both of which can enter the pentose phosphate pathway and be degraded [40]. Vitamin C autoxidizes easily to the superoxide anion (O_2_^•−^) and ascorbyl anion (ASC^•−^) in vitro (Figure 1) [45]. The degradation of vitamin C may be prevented by the phosphate moieties in the cell culture medium [46]. For example, the supplementation of 75 μM ascorbate with 500 μM 2-phosphoascorbate administration can maintain detectable levels of vitamin C for 72 h in the culture medium (1 ng/10^6^ cells) [47,48].

**Figure 1 ijms-25-03114-f001:**
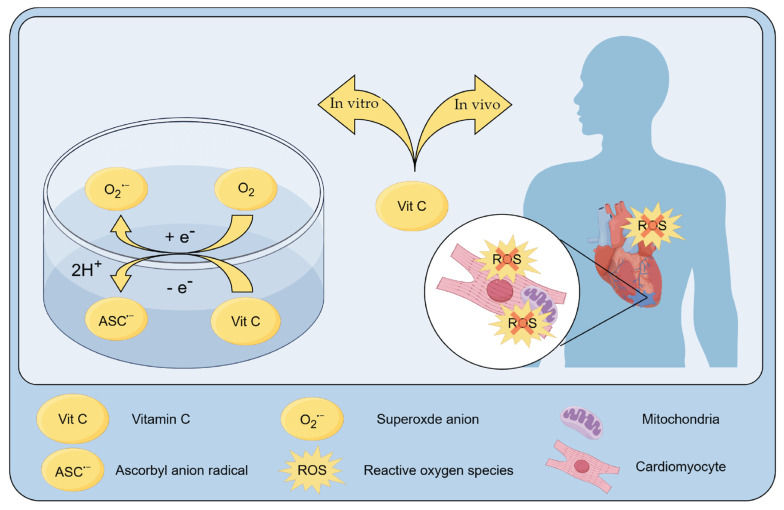
Metabolism of vitamin C in vivo vs. in vitro. In vivo, vitamin C is oxidized to dehydroascorbic acid (DHA) when reacts with ROS. In vitro, ascorbate autoxidizes easily to O_2_^•−^ and ASC^•−^.

## 3. The Role of Vitamin C in Cardiovascular Pathologies

The molecular regulators, such as ROS, are continuously generated in response to various conditions and play a key role in the pathophysiology of the heart. For example, an optimum heart function requires an ideal balance between ROS-producing enzymes and antioxidants, yet our knowledge remains limited in this regard [49]. Therefore, the increased production of ROS and oxidative stress are critical modulators in the healing and remodeling processes after MI [50,51].

### 3.1. Role of ROS in the Heart

The causative role of ROS in cell pathology remains unknown. ROS appear to be generated as degradation products during cellular metabolism and respiration and differ in their origin. Novel methodologies need to be developed to allow comprehensive investigations of the mechanism and regulation of ROS production at the cellular level.

Low levels of ROS (e.g., 10 μM H_2_O_2_, 500 nM H_2_O_2_) can be considered potent redox signaling messengers responsible for normal metabolic processes [52,53]. On the contrary, abnormal accumulation levels of ROS are associated with the loss of signaling and cellular damage [54]. Low levels of O_2_^•−^ and H_2_O_2_ can also increase the expression levels of hypoxia-inducible factor-1 (HIF-1) under hypoxia [54]. As a result, HIF-1 upregulates the transcription of genes responsible for angiogenesis, such as vascular endothelial growth factor [55]. Further, low levels of ROS play an important role in cardiomyocyte differentiation and proliferation [53]. However, chronically low levels of ROS are associated with prolonged cellular stress and cardiomyopathy [56].

Excessive ROS (1000 µM H_2_O_2_ [57], 100 µM H_2_O_2_ [58]) is reported to induce mutations of nuclear [59] and mitochondrial DNA [60], affecting the replication, proteins post-translation modifications [61], and membrane phospholipids [60] in the cardiac cells, accelerating aging and cell death [62].

Due to the higher diffusion rate of O_2_ in the myocardium, compared to other tissues, more ROS is produced under oxidative stress [63]. Indeed, in pathological conditions such as cardiac ischemia/reperfusion, cardiac hypertrophy, and heart failure, clinical data show the correlation of oxidative stress with an increased level of redox, and proinflammatory and profibrotic signaling [64,65].

Following ischemic injury, the reperfusion increases the expression of ROS, allowing large amounts of O_2_^•−^, H_2_O_2_, etc. to accumulate in the heart [66]. Concomitantly, under mechanical shear stress, a burst of both NO and O_2_^•−^ is released in the mitochondria, generating peroxynitrite and decreasing the bioavailability of NO. In a failing heart, a large pool of ROS production leads to an increase in mitochondrial permeability and outer membrane rupture, releasing apoptotic signaling molecules, such as cytochrome c (cyt c) [67].

Recent studies have revealed a strong similarity between the deleterious effects caused by the generation of ROS in myocardial tissue following ischemia/reperfusion and a specific iron-dependent form of non-apoptotic cell death known as ferroptosis [68]. The critical relationship between ROS and ferroptosis is particularly relevant in the context of MI, where both increased oxidative stress and iron overload coincide. Ferroptosis leads to cellular death through iron-catalyzed lipid peroxidation. This process can be prevented by an antioxidant, selenoprotein glutathione peroxidase 4 (GPX4), a vital enzyme that detoxifies lipid peroxides and stops ferroptosis [69,70]. Reliable therapeutic strategies, concerning the ischemic heart, could target ferroptosis to reduce reperfusion injury by the manipulation of iron availability or lipid peroxidation.

Additionally, the interaction between ROS, iron balance, and lipid peroxidation in the myocardium opens the way for new therapeutic targets in the management of ischemic heart diseases. For example, the investigation of molecules and pathways of ferroptosis, including ferroportin [71], FOXO1 [72], and others, may provide useful insights into the causal relationships between iron metabolism, ROS, and myocardial viability following ischemia.

The usage of H_2_O_2_ outside the cells may exceed 100–5000 times the usage in the cells (10 nM) [73] and, in the chronic inflammatory tissue, these values may be even ten times higher [73]. Enzymatic antioxidants, such as superoxide dismutase (SOD) and glutathione peroxidase (GPx), or non-enzymatic antioxidants, such as vitamin E and N-acetylcysteine, were reported to play important roles in scavenging ROS [63]. Copper zinc superoxide dismutase (Cu/Zn-SOD), located in the cytosol and the mitochondrial intermembrane space [74], and manganese superoxide dismutase (Mn-SOD), located in the mitochondrial matrix [75], catalyze O_2_^•−^ to H_2_O_2_ in the heart. Catalase (CAT) and GPx [76,77] are vital antioxidative enzymes, in both mitochondria and cytosol, that scavenge the excess H_2_O_2_ in the heart (Figure 2).

ROS, such as H_2_O_2_ and hydroxyl (OH^•−^), can be reduced by GSH, which is present in the cytosol, the nuclei, and the mitochondria [78] (Figure 2). Pathological conditions will tip the scale towards the failure of antioxidant clearance rather than the synthesis of abnormal amounts of ROS.

### 3.2. Cytosolic Sources of ROS (ctROS) in the Heart

At specific concentration levels, ROS has a detrimental or beneficial role in cell function (levels see Section 3.1) [79]. Oxidative stress is one of the main contributors to generating uncontrollably large amounts of ROS. For example, in the failing heart, oxidative stress is associated with significantly increased ctROS levels. ctROS signaling induces cardiomyocyte hypertrophy and cell death, influencing extracellular matrix remodeling and left ventricle chamber dilation [80,81]. It is important to address how ctROS signaling, from various sources, affects the heart, and how the heart finds reserves for the preservation of its function. For example, ctROS produced by NOX can lead to the activation of profibrotic genes such as NF-κB or matrix metalloproteinase (MMP)-2 [82] and, therefore, to increase cardiac fibrosis [83]. In the case of ischemia-reperfusion, the XO-associated abnormal ctROS synthesis leads to oxidative stress-mediated F-actin depolymerization, which induces mitochondrial dysfunction and fission, and apoptosis of the cardiac microvascular endothelial cells [84,85].

The major source of ctROS is the seven-NOX enzymatic complexes. In particular, NOX2 and NOX4 isoforms seem to have distinct physiological and pathophysiological roles in the heart [51,86]. For example, angiotensin II induces the activation of cytosolic subunits of NOX2 such as p47phox, p67phox, p40phox, and Rac1. These activated subunits bind to flavocytochrome and induce the synthesis of radical species such as O_2_^•−^ and OH^•−^ [87]. These reactive species stimulate calcium release from the sarcoplasmic reticulum in myocytes and excitation–contraction coupling [87], which can result in contractile dysfunction and myocyte death. Activated NOX2 can also modulate interstitial fibrosis via activating metalloproteinase activity following MI [88]. Although NOX4 activity is controversial, it has an important role in angiogenesis [89], mediating metabolic stress responses [90] and detoxifying responses [91]. NOX4, localized in the endoplasmic reticulum and mitochondria, generates large amounts of detrimental H_2_O_2_ [92]. Another cytosolic source of ROS is H_2_O_2_, which may derive from XO and uric acid [93].

Another important source of cytosolic ROS in myocytes is the uncoupling of endothelial nitric oxide synthase (eNOS) [94] and inducible nitric oxide synthase (iNOS) [95]. Their upregulation is associated with increased infarct size and cardiac dysfunction in ischemia/reperfusion injury in the diabetic rat heart [95] and chronic pressure-induced hypertrophy in wild-type mice, respectively [94].

Although cardiac cytochrome P450 enzymes have major cellular metabolic and homeostasis characteristics, they also represent a rich cytosolic source of ROS, as observed in cigarette smoking patients [96]. The inhibition of cardiac cytochrome P450 enzyme, as demonstrated in a rat heart model, may constitute a potential clinical approach to suppressing ctROS formation [97].

Extracellularly, a minor potential source of H_2_O_2_ production is the autoxidation of small molecules such as catecholamines [98]. In vivo experiments show that catecholamines-derived ROS cause oxidation of lipids in the rat heart and thus are associated with negative cardiac remodeling [99,100].

Among ctROS sources, NOX2 seems to be the most important in the pathological heart. A high level and activity of NOX2 produces large amounts of ROS, causing injury and initiating the fibrotic progress of the pathological condition, such as ischemia.

### 3.3. Mitochondrial Sources of ROS (mtROS) in Heart

Mitochondria are the “energy-producing motor” of cardiac cells, as the heart constantly demands the synthesis of high levels of ATP. The ATP synthesis undergoes a multistep process as follows: post-translational modifications (e.g., oxidative phosphorylation); electron transfer across the five protein complexes (complexes I–V); and two transporters (coenzyme Q and cyt c). Oxidative stress limits ATP synthesis [101], resulting in uncontrolled mtROS formation and accumulation, thus impairing the contractile function of cardiac cells [102,103]. Following the Krebs cycle, the oxidation of metabolic products leads to the formation of mtROS in the form of O_2_^•−^ [63] and to reduced nicotinamide-adenine dinucleotide (NADH) [104] or reduced flavin adenine dinucleotide (FADH_2_) [105] (Figure 2). As a result of electron transport chain (ETC) across complexes I and III, mtROS production associates partially to O_2_ → O_2_^•−^, commonly described as the main sources of mtROS [106,107]. However, enzymes in the respiratory complex II can also reduce O_2_ levels under hypoxia settings, thereby impeding ATP production [108]. Several research findings and examples that explain the involvement of complexes I–V in regulating the remodeling of myocardial following injury are described below. Key research questions are identified and systematically reviewed such as: (i) what are the key factors responsible for the dysregulation of mtROS production; (ii) what are the effects of mtROS on the function of mitochondria and cardiac cells; and (iii) what is the role of mtROS in the progress of MI?

Complex I (NADH dehydrogenase) is the first enzyme of the mitochondria respiratory chain. Complex I is the most sensitive respiratory chain complex, being identified as the central site of ROS production in the mitochondria. Structurally, complex I has a hydrophobic unit immersed in the inner membrane and a hydrophilic unit faced in the matrix (Figure 2A). The hydrophobic unit of complex I, under critical thermodynamic conditions, favors the reverse electron transport and reduction of ubiquinone, dismutase of O_2_ to H_2_O_2_, and OH^•−^ species formation. The hydrophilic unit participates in pumping H^+^ into the intermembrane space and mediates the transfer of high energy electrons from NADH to flavin mononucleotide to form the NAD^+^, then to coenzyme Q (ubiquinone) to form ubiquinol (QH_2_), supported by the Fe-S carrier [109] (Figure 2A). Both flavin mononucleotide (FMN)- and Q- binding sites may be subjected to electron leakage from ETC under pathological conditions, leading to the formation of O_2_^•−^ species [110,111]. At least 40% of all mitochondrial disorders are associated directly with DNA mutations in subunits of complex I, because elevated levels of O_2_^•−^ cannot be optimally regulated by SOD, CAT, and GPx [112] (Figure 2A). A rotenone toxin model revealed the increased release of mitochondrial O_2_^•−^ in the A7r5 SMCs derived from rat aorta [113]. Presumably, a fraction of O_2_^•−^ is dismutated to H_2_O_2_ by SOD and further reduced to H_2_O by antioxidants, such as CAT and GPx in the mitochondrial matrix [112]. Excessive mitochondrial H_2_O_2_ causes direct damage to the internal structures, including proteins, lipids, and DNA in the mitochondria of cardiomyocytes [114]. Another fraction of O_2_^•−^ binds to NO and forms peroxynitrite, which may result in the upstream of rotenone binding sites and a progressive shift of redox potential [115], leading to the decrease of ATP production in the mitochondria and the dysfunction of cardiomyocytes [116].

Using the artificial acceptor hexaammineruthenium (III) reductase, Stepanova, A. et al. revealed that increased levels of covalently non-bound flavin mononucleotide and O_2_ depletion lead to reduced mitochondrial complex I activity and ATP content [117]. Moreover, to the best of our knowledge, the complex metabolic interplay among, e.g., fatty acids, glycero-3-dehydrogenase, electron-transferring flavoprotein, etc. and the mechanisms of mitochondrial O_2_^•−^ release are yet not known in vivo. However, the redox potential shift evidenced either in the in vitro model [118] or in the acute brain ischemia/reperfusion mice model [119,120] can be associated with changes in cardiac function.

In contrast to complex I, in cardiac complex II (succinate dehydrogenase) in ischemia/reperfusion injury, higher mitochondrial oxygen consumption is detected [121]. In complex II, less H^+^ is generated and less ATP is formed from FADH_2_ than from NADH. In the presence of succinate dehydrogenase, succinate converts into fumarate and reduces FAD to FADH_2_, which donates electrons to the Fe-S clusters, to ultimately be transferred to the reduced ubiquinol form of coenzyme Q [122] (Figure 2B). The dicarboxylate-binding site of complex II (II_f_ site) is associated with electron leakage from ETC and ROS generation [123,124] (Figure 2B). In vitro research using isolated mitochondria suggested that the II_f_ sites were reduced after the inhibition of complex II by atpenin A5 in myocardial ischemia [123]. As a result, the reduced fraction of site II_f_ is inversely associated with ROS production in complex II, thereby impairing the mitochondrial activity and cardiac cell function [123]. Succinate is an important substrate of complex II that accumulates and is rapidly oxidized by complex II upon ischemia/reperfusion, thus leading to the accumulation of large amounts of O_2_^•−^ in complex II [50,120]. Interestingly, at the tissue level, malonate, the competitive inhibitor of complex II, significantly reduces mtROS levels and infarct size in the heart after ischemia/reperfusion [125] by increasing the mitochondrial oxygen consumption rate [121]. In this context, complex II might serve as a metabolic target source within the respiratory chain.

Complex III (cyt c reductase) catalyzes the transfer of electrons from the readily formed ubiquinol to cyt c in the Q cycle [126] via two distinct redox paths [127,128], while H^+^ is pumped back into the intermembrane space [126] (Figure 2C). One path uses the Fe-S cluster group and heme group of cyt c_1_ to receive four high-energy electrons from two molecules of QH_2_. Two of the electrons are transferred to the water-soluble molecule of cyt c, and another two electrons are transferred to cytochrome b to participate in the second electron transfer path [128]. The second electron transfer path uses cyt b as electron—cargo to partially reduce the Q to a semiquinone radical anion (Q^•−^). Further, Q^•−^ receives another electron from cyt b, generating QH_2_ at the end of the electron transfer chain [106,127]. In vitro, the decreased activity of mitochondrial complex III from aging rat hearts is associated directly with increased ROS production, thereby correlating to the increased infarction size after ischemia/reperfusion injury [129]. As a possible mechanism, aging increases the ionic strength around mitochondrial complex III, which decreases the activity of subunits VIII or X in the cyt c binding site of complex III [130]. Likewise, the mitochondrial complex III activity is reduced significantly in the ischemia/reperfusion rabbit model, which is associated with elevated ROS levels [123]. Similarly, studies in rats revealed that ischemia/reperfusion destroys the structures of hemes b, c, and c_1_, and thereby the activity of complex III in cardiac cells [131]. To regulate ROS production in the post-ischemic heart, the large quantities of O_2_^•−^ accumulated in the reduction sites of Qo and Qi of complex III should be reduced [132]. The reduced O_2_^•−^ synthesis in the Q sites of cyt c can be achieved pharmacologically by inhibiting the Qo and Qi sites with antimycin A [133] and myxothiazol [134], e.g., in rat hippocampal CA1 cells, with stigmatellin, e.g., in cremaster arterioles of wild type mice [135], and with Ginsenoside Rc, e.g., in rat cardiomyocytes after ischemia/reperfusion [136].

The complex IV (cyt c oxidase) transfers the electrons from reduced cyt c molecules to oxygen to generate H_2_O molecules [137]. Following the ETC, a H^+^ electrochemical gradient is established across the inner mitochondrial membrane [137] (Figure 2D). Cuprous ion (Cu^+^) is able to react with O_2_ and form a peroxide bridge between heme groups and cupric ion (Cu^2+^); thus, the electrons from cyt c are transferred to the heme groups [138]. The scission of the peroxide bridge occurs as a result of H^+^ extraction from the mitochondrial matrix and the reduction of Cu^2+^ to Cu^+^. H_2_O molecules are generated as reaction products [138]. The heme a_3_ site is the O_2_ binding site in complex IV (Figure 2D). This site can be competitively inhibited by molecules such as H_2_S, CO, and NO [139,140,141], thus blocking the electron transport from cyt c to O_2_ and generating increased levels of ROS [142]. The insufficient O_2_ supply during MI results in increased electron leakage, leading to the impairment of complex IV activity and the accumulation of large amounts of mROS [143,144]. As such, the overproduction of mROS reduces the oxidant scavenging capacity of mitochondrial antioxidants, such as MnSOD [75], Cu/ZnSOD [74], and GSH [78].

In the in vivo MI models, the increased levels of ROS (OH^•−^ and O_2_^•−^) are associated with 30% to 50% activity of some subunits of complex I (e.g., ND1), such as cyt b in complex III and the whole IV, thereby explaining the impaired activity of these complexes [143,144,145].

Recently, it has been proved that the hyperphosphorylation of mitochondrial proteins in complex IV in myocardial cells is induced by activating protein kinase A (PKA) signaling pathways, which alters the function of mitochondrial proteins [143] and impairs mitochondrial activity [143]. Cyclic adenosine monophosphate (cAMP) in mitochondria is an important activator of PKA [146], causing a decrease in the complex IV activity of cardiac cells after ischemia/reperfusion injury [143,147] (Figure 2D).

Complex V (ATP synthase) uses the motor force of H^+^ ions, established in the forehead redox sequences, to synthesize ATP molecules by ADP or phosphate molecules, releasing them into the mitochondrial matrix [148] (Figure 2E). Although complex V does not generate ROS, under pathological conditions, the mitochondrial permeability transition pore (MPTP) is activated by the presence of a large accumulation of ROS synthesized by previous complexes. The main components of MPTP are the voltage-dependent anion channel and the adenine nucleotide translocator. Several other MPTP components, e.g., cyclophilin D may interact with the adenine nucleotide translocator, whereas complex V dimers bind to cyclophilin D from the mitochondrial matrix [149] (Figure 2E).

Also, the phosphorylation of complex V dimers subunit c leads to the opening of MPTP [150], subsequently uncoupling oxidative phosphorylation and the production of mtROS [151]. The opening of MPTP may also induce cell apoptosis or necrosis by releasing pro-apoptotic proteins, such as apoptosis regulator BAX [50,120] (Figure 2E), thus increasing the infarct size after ischemia/reperfusion [152]. In the in vivo animal experiments using dogs with microembolization-induced heart failure, the improved activity or expression of mitochondrial complex V is associated with a preserved ejection fraction of the heart [153].

**Figure 2 ijms-25-03114-f002:**
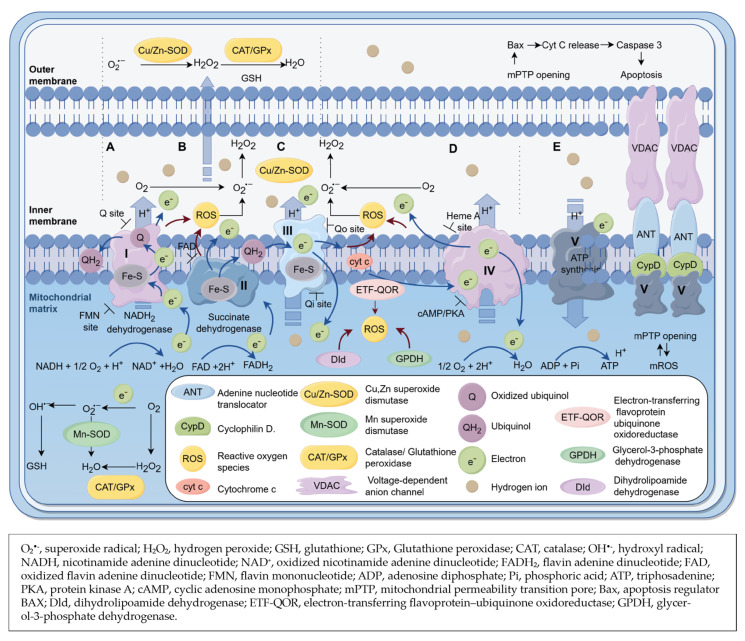
Schematic diagram of various sources of ROS generation and its effect on mitochondrial activity in CVD (brown arrows point to the enzymatic sources of ROS formation). Oxidized ubiquinol receives two electrons from Fe-S clusters of complexes I (**A**) [154] and II (**B**) [122] to form ubiquinol; ubiquinol transports electrons to Fe-S clusters in complex III. Cyt c mediates the transport of electrons from complex III (**C**) to complex IV (**D**) [126]. Complex IV delivers the electrons to O_2_ and forms H_2_O. Hydrogen ions are pumped by complexes I [154], III [126], and IV [137] from the mitochondrial matrix to the mitochondrial intermembrane space to generate the H^+^ electrochemical gradient for ATP generation. MnSOD [75], Cu/Zn-SOD [74], CAT [76], GPx [76], TRx [77], and GSH are vital for the scavenging of ROS in the mitochondria. Cu/Zn-SOD, CAT, GPx, thioredoxins, and GSH also play important roles in scavenging ROS in the cytosol [74,76,78,155]. Complex V (ATP synthase) synthesizes ATP molecules [148] (**E**). Complex V dimers participate in the formation of MPTP, which also includes VDAC, ANT, and CypD. The overproduction of ROS in the mitochondria causes the opening of MPTP, producing a large amount of mtROS [156]. The opening of MPTP can even activate Bax and then lead to the release of cyt c, the activation of caspase 3 [50,120], and apoptotic cell death [152,157] (**E**).

Among the various sources of ROS in the heart, mitochondrial complexes I and III are the key players in cardiac pathology. They produce a large amount of ROS, which induces damage to the cellular structures (e.g., the damage to DNA, proteins, and lipids) and even the death of the cardiomyocytes, particularly in diabetes [158]. In conclusion, ROS, which are mainly produced by mitochondrial complexes I and III in cardiomyocytes, play an important role in cardiac pathology.

### 3.4. Role of Vitamin C as a Scavenger of ROS

Vitamin C is the most known antioxidant that scavenges various types of radicals, e.g., OH^•−^, H_2_O_2_, and O_2_^•−^ [159,160,161]. Vitamin C either donates one electron to O_2_^•−^ to produce ASC^•−^ or loses a second electron to form its oxidized form, DHA [159,160,161] (Figure 3A–E). Observational studies demonstrated the antioxidant capacity of vitamin C even when antioxidative enzymes, such as SOD and CAT, are decreased [162,163]. Further, the dietary supplementation of 1000 mg/day for 6 weeks, in the case of stroke-prone spontaneously hypertensive rats, downregulated the expression of O_2_^•−^ in association with the reduced activity of NOX [164].

Interestingly, vitamin C increases the activity of antioxidative enzymes. For example, small supplementation amounts of 30 μM of vitamin C significantly reduced ROS amounts within 60 min in the isolated mitochondria [165] by increasing the activity of Mn-SOD and GPx [165] (Figure 3A–D).

As we have mentioned in Section 3.1, CAT and Cu/Zn- SOD are key antioxidative enzyme regulators of mitochondrial function and they can be stimulated by vitamin C. An in vivo experiment in diabetic rats showed that the supplementation of vitamin C (50 mg/kg per day for 4 weeks) increased the activity of CAT and Cu/Zn- SOD in the liver [166].

Glutathione (GSH) is a well-known nonenzymic antioxidant with a protective role in the cells. It can be decreased by oxidative stress in in vivo MI animal models [167], whereas its oxidized form (GSSG) is increased in isoproterenol-induced cardiac remodeling, which is also related to oxidative stress [168]. Vitamin C maintains cellular GSH levels within different sites [169,170] (Figure 3A–D). It is worth noting that a supplementation of vitamin C of 200 mg/day for patients aged 33–63 years with Type 2 diabetes was sufficient to significantly increase the activity of SOD by 3% and GPx by 52% in blood samples [171].

Vitamin C (1 μM) also prevents lipid peroxidation by clearing ROS and maintaining the levels of antioxidants [172,173]. Thus, the development of ferroptosis is inhibited [172], allowing us to speculate that the damage ferroptosis causes to cardiomyocytes after MI may be inhibited by vitamin C.

Moreover, vitamin C has an antioxidant role in the treatment and prognosis of MI in animal models. Pigs that received a vitamin C hydrogel after MI surgery showed improved cardiac function, increased vascular density, and reduced fibrosis compared to the untreated group. This improvement was attributed to the suppression of oxidative stress by vitamin C [174]. In another study, MI was induced in rats with isoproterenol, leading to enhanced oxidative stress and elevated levels of inflammatory factors such as tumor necrosis factor-α. The supplementation of vitamin C (250 mg/kg body weight, intragastric injection) reduced the oxidative stress, inflammation, and size of the infarction in the heart [175]. Likewise, under MI conditions, the oral supplementation of vitamin C (250 mg/kg body weight) in rats further confirmed its antioxidant role by preserving cardiac function [176]. This aligns with recent clinical evidence suggesting vitamin C’s potential in reducing oxidative stress and improving outcomes for patients with acute coronary syndromes and those undergoing cardiac surgeries [177].

Although the mechanisms for reducing the cardiovascular risk remain unclear, the role of vitamin C in scavenging ROS [178], enhancing NO production [179], and its potential benefits for microperfusion [177], myocardial protection [79], and hemodynamic stability [180]—especially in critically ill patients [181]—is increasingly recognized.

Thus, despite being controversial, preclinical and clinical studies have demonstrated that vitamin C plays beneficial roles in cardiovascular diseases (either in peri-operative or post-operative interventions) [182]. However, the effects of vitamin C seemed to be dose- and time-dependent. For example, overdoses of vitamin C (≥300 mg/day, Table 1) may increase cardiovascular mortality [183]. This highlights the importance of careful dosage management in therapeutic settings. Additionally, a meta-analysis has linked vitamin C intake levels with an elevated risk of developing kidney stones [184].

In conclusion, vitamin C can directly decrease ROS levels and protect the cardiovascular system from oxidative stress by improving the electron transport chain mitochondrial function. Vitamin C also scavenges mtROS in the cardiovascular system by maintaining the levels of intracellular antioxidants, including Mn-SOD, GPx, CAT, Cu/Zn-SOD, and glutathione. Although vitamin C may be considered a potential effector in disease management strategy, care should be taken concerning its dose and time-dependent administration, particularly at dosages exceeding 300 mg/day.

**Figure 3 ijms-25-03114-f003:**
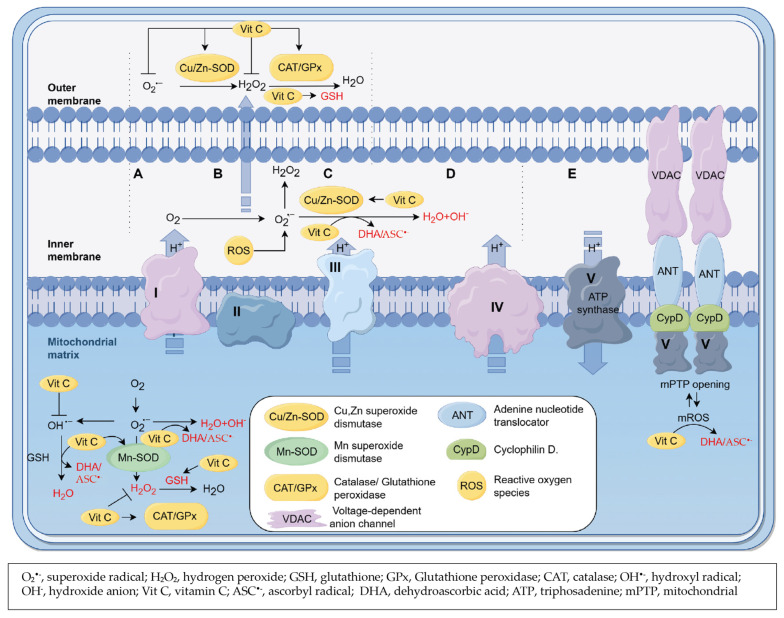
Overview of the effect of vitamin C on scavenging mitochondria-produced ROS in CVD. Vitamin C in the mitochondria reduces O_2_^•−^ [159], OH^•−^ [78], and H_2_O_2_ [78] produced by mitochondrial complexes I (**A**)**,** II (**B**), III (**C**), IV (**D**), and V (**E**), and is oxidized to DHA or ASC^•−^. Vitamin C is also reported to increase the activity or levels of GSH [78], Mn-SOD [165,185], GPx [165], CAT [166,186], Cu/Zn-SOD [185,186] (**A**–**D**), and therefore enhance the ability to scavenge ROS in the mitochondria in the CVD condition.

**Table 1 ijms-25-03114-t001:** The outcome of vitamin C administration in clinical trials of cardiovascular pathology. PCI, percutaneous coronary intervention; MI, myocardial infarction; HF, heart failure; iv, intravenous; SOD, superoxide dismutase; XO, xanthine oxidase.

Population	Study Size (n)	Age (Years)	Intervention (Doses of Vitamin C)	Trial Duration	Outcomes	Reference
Male with the first MI	180	<65	Habitual	3 months	Low plasma concentration had no benefit on MI	[35]
Patients with sepsis and ARDS present for less than 24 h	167		50 mg/kg in dextrose 5% in water, n = 84 (iv)	96 h	No improvement of organ dysfunction sores, inflammation and vascular injury	[36]
Male with no prevalent HF	3919	60–79	Habitual	11 years	High plasma concentration reduced the risk of HF with and without myocardial infarction	[39]
Postmenopausal women with diabetes	1923	55–69	≥300 mg/d (orally)	a mean time of 15 years	Higher cardiovascular mortality	[183]
Healthy adults	13421	a mean of 61	320–1110 mg/d (orally)	a mean time of 11 years	Lower cardiovascular mortality	[187]
Patients with clinically stable class I or II effort angina pectoris	56	a mean of 67 (50–84) years	16.6 mg/min over 1 h before PCI (iv)	-	Microcirculatory reperfusion improved; Oxidative stress decreased	[188]
Patients with asymptomatic adults with elevated coronary calcium scores	1005	50–70	Combination of vitamin C 1 g, vitamin E 1000 U and atorvastatin 20 mg daily (orally)	a mean duration of 4.3 years	Levels of total cholesterol, low-density lipoprotein cholesterol and triglycerides reduced	[189]
Patients with MI	800	A mean of 62	1000 mg in 12 h (iv) followed by 1200 mg/day combined with vitamin E 600 mg/day (orally)	For 30 days	In-hospital cardiac mortality, non-fatal new myocardial infarction, shock/pulmonary edema, etc. decreased.	[190]
Adults without MI or stroke	14641	≥50	500 mg/day (orally)	For 8 years	No benefit on MI, stroke or cardiovascular mortality.	[191]
Patients with MI	84	53.1 ± 11.2 in the antioxidant treatment group, 55.1 ± 8.4 years in the control group	250 mL 20% mannitol (iv) over the first hour, 1000 mg of vitamin C in 500 mL 5% glucose (iv) over the first 4 h	-	Short term (e.g., cardiogenic shock), and long term (e.g., left ventricular insufficiency) complications decreased.	[192]
Patients with MI	120	-	500 mg, twice per day after MI (orally)	5 days	Levels of SOD and total thiols increased Levels of XO and malondialdehyde decreased.	[193]
Male with MI	21	60.5 ± 6.8	2000 mg after MI (orally)	-	Oxidative stress decreased.	[194]

## 4. Conclusions

The beneficial role of vitamin C in the cardiovascular system remains controversial, despite solid evidence demonstrating that vitamin C reduces excessive ROS formation. The non-linearity between vitamin C supplementation levels and variable ROS sources seems to shift the oxidative-redox events in the cardiac cell, which might impair mitochondrial function, highlighting the importance of further studies at the molecular level. The controversial effects of vitamin C on cell culture do not translate into side effects in clinical studies, thus polarizing the research outcome and making it difficult to draw clinical guidelines for vitamin C implementation as a therapeutic medicine. Although vitamin C demonstrates a strong antioxidant function, at both the cellular and organ levels, further studies are needed to provide cumulative time-dependent dose data to clarify the role of vitamin C in the prevention and/or cure of cardiovascular diseases. The controversial risks of vitamin C administration in human studies, particularly in terms of kidney stone formation at chronically high doses, underscore the necessity of developing future combination therapies for cardiovascular diseases.

An interesting future strategy concerns the establishment of an effective role of vitamin C in ferroptosis, since the interaction between ROS, iron balance, and lipid peroxidation in the myocardium is critical for regulating the fate of cells. This may provide useful insights into new therapeutic targets for the management of ischemic heart diseases.

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
