# Peer review of "Vitamin C as Scavenger of Reactive Oxygen Species during Healing after Myocardial Infarction"

_ijms, 2024, doi:10.3390/ijms25063114_

Round 1
Reviewer 1 Report
Comments and Suggestions for Authors
The review entitled 'Vitamin C as scavenger of ROS during healing after myocardial infarction' discusses the clinical application of Vitamin C as an adjunct therapy in the management of cardiovascular disorders. Review is interesting, but it has many flaws. Here are my comments:
-The language throughout the review is very unscientific. Here are some examples. These sentences must be modified to convey a clear message: 'Modern management mechanical (balloon angioplasty) and pharmacological (thrombolysis) therapies have shown the restored blood flow effect in patients with MI'' '. However, the invasive measures next to the recruitment and activation of inflammatory cells may lead to reperfusion injury, which is a burst of free radical formation';' Therefore, management strategies to preserve the onset of physiological values of cardiac function are imperatively needed.'; 'The energy reservoir produced by mitochondrial activity fuels daily the heart with about 6 kg ATP to impose 100,000 times beat and to pump approximately 10 tons of blood through body'; 'Antioxidant treatments have been proposed to alleviate these harmful mechanisms by enhancing the antioxidant defense systems via using numerous antioxidants such as vitamin C [13], vitamin E [14],'.
-'dehydroascorbic acid (DHA) by oxidants, such as hydrogen peroxide (H2O2), or by vitamin C oxidase'; the latter is an enzyme
-Message in Fig. 1 is not clear: what is meant by superoxide radical and ASC radical and how are they generated? Further, how Vitamin C affects the CV system should be shown very clearly.
-'The formation of a ROS pool uses cytosolic sources (nicotinamide adenine dinucleotide phosphate oxidase family (Nox)', Nox is located in the plasma membrane, so it is incorrect statement.
-iNOS and eNOS (under certain circumstances in CV system) need to be included as important sources of ROS.
-In Fig. 2, enzymatic sources responsible for ROS generation in the CV system need to be shown.
-Fig. 3 discusses about so many antioxidants; its focus should be vitamin C and its consumption by different oxidants and its regeneration by other enzymatic or non-enzymatic antioxidants. All other biochemical moieties should be removed. Fig. needs to be simplified.
-Table detailing the advantages and disadvantages of vitamin C supplementation in CV patients in different clinical trials needs to be included.
-Animal studies on MI conducted with vitamin C need to be included.
Comments on the Quality of English LanguageIt is difficult to understand the text.
Author Response
The review entitled 'Vitamin C as scavenger of ROS during healing after myocardial infarction' discusses the clinical application of Vitamin C as an adjunct therapy in the management of cardiovascular disorders. Review is interesting, but it has many flaws.
ANSWER: We thank very much that you find our review interesting, and we are grateful for your suggestions to improve our manuscript.
Here are my comments:
- The language throughout the review is very unscientific. Here are some examples. These sentences must be modified to convey a clear message: 'Modern management mechanical (balloon angioplasty) and pharmacological (thrombolysis) therapies have shown the restored blood flow effect in patients with MI'' '. However, the invasive measures next to the recruitment and activation of inflammatory cells may lead to reperfusion injury, which is a burst of free radical formation';' Therefore, management strategies to preserve the onset of physiological values of cardiac function are imperatively needed.'; 'The energy reservoir produced by mitochondrial activity fuels daily the heart with about 6 kg ATP to impose 100,000 times beat and to pump approximately 10 tons of blood through body'; 'Antioxidant treatments have been proposed to alleviate these harmful mechanisms by enhancing the antioxidant defense systems via using numerous antioxidants such as vitamin C [13], vitamin E [14],'.
ANSWER: We are grateful for pointing this out, we have now corrected the phrases throughout all the manuscript with the help of native English-speaking college.
- 'dehydroascorbic acid (DHA) by oxidants, such as hydrogen peroxide (H2O2), or by vitamin C oxidase'; the latter is an enzyme
ANSWER: We have now corrected the phrase.
- Message in Fig. 1 is not clear: what is meant by superoxide radical and ASC radical and how are they generated? Further, how Vitamin C affects the CV system should be shown very clearly.
ANSWER: We have now made clearer the message in the Figure 1. The effects of Vitamin C on CV system, as resulted from animal and human studies, are stated now in the revised chapter 3.4.
- 'The formation of a ROS pool uses cytosolic sources (nicotinamide adenine dinucleotide phosphate oxidase family (Nox)', Nox is located in the plasma membrane, so it is incorrect statement.
ANSWER: We have now deleted this phrase from the manuscript, since it was not essential for the message and reformulate the information regarding the Nox (see reviewed chapter 3.2.)
- iNOS and eNOS (under certain circumstances in CV system) need to be included as important sources of ROS.
ANSWER: We are grateful for pointing this out. We have now discussed iNOS and eNOS as important sources of ROS in pathological conditions in the heart (see revised chapter 3.2.).
- In Fig. 2, enzymatic sources responsible for ROS generation in the CV system need to be shown.
ANSWER: We thank for the suggestion. We have now added the representative enzymes as sources of ROS in the Figure 2. The representative sources of enzymes are pointed by brown arrows towards ROS, in Figure 2.
- Fig. 3 discusses about so many antioxidants; its focus should be vitamin C and its consumption by different oxidants and its regeneration by other enzymatic or non-enzymatic antioxidants. All other biochemical moieties should be removed. Fig. needs to be simplified.
ANSWER: We have now simplified the Figure 3 and removed all the biochemical information, focusing on the metabolism and intervention of vitamin C on mitochondria and ROS trafficking.
- Table detailing the advantages and disadvantages of vitamin C supplementation in CV patients in different clinical trials needs to be included.
ANSWER: We are grateful for this suggestion, we have now included all information about known clinical trials using vitamin C, as well as their results on cardiac outcome in the table 1.
- Animal studies on MI conducted with vitamin C need to be included.
ANSWER: We have now mentioned and discuss the animal studied on MI conducted with vitamin C, as suggested. This can be now found in the section 3.4 of the revised manuscript.
Reviewer 2 Report
Comments and Suggestions for Authors
Major Comments:
On page 4, paragraph 2, line 3: the reference 46 is not the correct one for the given statement.
In Figure 2 abbreviations: OH· is hydroxyl radical not hydroxide anion.
On page 10, paragraph 1, line 3: vitamin c donates two electrons to form its oxidized form (DHA).
On page 10, paragraph 3: “chapter 1.2” There is no chapter 1.2 in the manuscript.
On page 10, paragraph 5: “chapter 3.1” There is no chapter 3.1 in the manuscript.
Author Response
On page 4, paragraph 2, line 3: the reference 46 is not the correct one for the given statement.
ANSWER: We thank very much from this observation; we have now corrected the statement according to the reference 46.
In Figure 2 abbreviations: OH· is hydroxyl radical not hydroxide anion.
ANSWER: We have now corrected the pointed-out mistake.
On page 10, paragraph 1, line 3: vitamin c donates two electrons to form its oxidized form (DHA).
ANSWER: We have now corrected and rephrase this information.
On page 10, paragraph 3: “chapter 1.2” There is no chapter 1.2 in the manuscript. On page 10, paragraph 5: “chapter 3.1” There is no chapter 3.1 in the manuscript.
ANSWER: We thank very much for pointing this out. We have now corrected the numbering of the chapters in the manuscript.
Reviewer 3 Report
Comments and Suggestions for Authors
The article is well-written and provides useful information on the role of vitamin C as an antioxidant during healing after MI, revealing its mechanisms.
All abbreviations should be defined at their first occurrence in text ex (ATP line 52, ADP line 53, ROS line 96)
One suggestion should be to discuss whether any side effects associated with vitamin C use have been found.
Are there studies of human subjects with MI who received vitamin C? If so, which were the doses used?
Author Response
The article is well-written and provides useful information on the role of vitamin C as an antioxidant during healing after MI, revealing its mechanisms.
ANSWER: We thank you this referee for considering our manuscript well written and useful. We have now addressed the indicated concerns and added the requested information to our revised manuscript.
All abbreviations should be defined at their first occurrence in text ex (ATP line 52, ADP line 53, ROS line 96)
ANSWER: We added now defined all the abbreviations when they first appear in the text.
One suggestion should be to discuss whether any side effects associated with vitamin C use have been found.
ANSWER: We have now added the findings regarding side effects associated vitamin C administration both in the chapter 3.4, lines from 491 to 510 and chapter 4, lines 530 to 540.
Are there studies of human subjects with MI who received vitamin C? If so, which were the doses used?
ANSWER: The doses used in the human studies are now included in the table 1, as well as the effects of the use of vitamin C on cardiac pathologies tested in the respective studies.
Round 2
Reviewer 1 Report
Comments and Suggestions for Authors
Previous comments not duly addressed by the authors
Comments on the Quality of English LanguageEnglish editing required
Author Response
We are very sorry that the referee considered that we couldn't response adequately to his requests. We think it is in part due to the English used in the manuscript. Therefore, we have now let the manuscript be revised by an English native-speaker, who improved significantly the understanding and made the manuscript much easier to be read (now mentioned in acknowledgement).